# PARAMETER-EFFICIENT ATTENTION TRANSFER FOR MULTI-MODAL TEST-TIME ADAPTATION

## ABSTRACT

Test-time adaptation (TTA) has proven effective in enhancing model robustness against unforeseen distribution shifts during testing. However, current TTA methods struggle when applied to multi-modal models. In this paper, we explore multi-modal TTA and reveal two key limitations of existing approaches: i) difficulty in mitigating attention shifts when dealing with biased modalities, and ii) insufficient exploitation of the synergy and complementarity among multiple modalities. To address these challenges, we propose a novel method called **P**arameter-**E**fficient **A**ttention **T**ransfer (PEAT), which strikes a balance between performance and efficiency. Specifically, we first discuss the modulation strategies for updating various model parameters and propose to adapt the self-attention modules. Furthermore, we design a modality-aware low-rank adaptation method to dynamically learn cross-domain attention patterns. Our approach introduces intra-modal and inter-modal interactions for LoRAs, where the former captures uni-modal domain information through modality-specific parameters, while the latter promotes cross-modal feature alignment in a unified space through modality-shared parameters. Extensive experiments conducted across various distribution-shifted modalities, including video, image, audio, and text, demonstrate that PEAT consistently outperforms existing state-of-the-art methods.

## 1 INTRODUCTION

In recent years, multi-modal learning has demonstrated significant advancements against cross-modal heterogeneity (Baltrušaitis et al., 2019; Bao et al., 2022; Wang et al., 2023; Guo et al., 2024b). In this context, multi-modal pre-training has emerged as a promising avenue for associating multiple communicative modalities, allowing for enhanced comprehension and performance in the various real-world downstream tasks, particularly in the audio-visual (Arandjelović & Zisserman, 2018; Gong et al., 2023) and vision-language (Radford et al., 2021; Li et al., 2022) domains.

In the pre-training paradigm, multi-modal pre-trained models can be adapted to specific domains through fine-tuning. This paradigm exhibits remarkable performance in the assumption of the known and fixed test domain. However, such a mild assumption is often violated in non-stationary and changing practical environments. In open-world scenarios, test samples may encounter natural variations or corruptions (*i.e.*, *distribution shifts*), which can be attributed to unpredictable factors such as weather changes, sensor degradation, etc (Hendrycks & Dietterich, 2019). Although multi-modal data boast rich and comprehensive information representation, the multi-modal models still suffer significant performance degradation against test-time distribution shifts.

Recently, fully test-time adaptation (TTA) methods propose to adapt the model using unlabeled samples during testing, which have been shown to boost robustness against distribution shifts in the test domain. Toward this goal, existing works focus on mitigating the *covariance shifts* across domains via normalization statistics adaptation. Specifically, the affine parameters are optimized in each test batch with self-supervised or unsupervised objectives, including, but not limited to, entropy minimization (Wang et al., 2021; Niu et al., 2022; 2023), pseudo labeling (Liang et al., 2020; Wang et al., 2022), and consistent regularization (Zhang et al., 2022).

However, these methods are constrained to uni-modal tasks, exhibiting suboptimal improvements when applied to multi-modal models. Therefore, it is worth exploring *how to bridge the gap between uni-modal and multi-modal adaptation*. For this issue, we rethink the limitations of existing

Table 1: Qualitative comparison of adapting various modulation parameters on VGGSound-C dataset with corrupted audio modality (severity level 5) regarding **Accuracy (%, ↑)**. "LN", "MLP", "SAF" and "Attn." denote the layer normalization, multilayer perceptron, self-adaptive attention-based fusion, and self-attention, respectively. △ (%, ↑) represents the average improvement in model accuracy compared with the source model. The **bold** number indicates the best result.

| Method | Noise | | | Weather | | | Avg. | △ | Param. |
|--------|-------|--------|--------|------|--------|------|------|------|--------|
| | Gauss. | Traff. | Crowd. | Rain | Thund. | Wind | | | |
| Source | 37.2 | 21.2 | 16.8 | 21.6 | 27.3 | 25.5 | 24.9 | - | - |
| ● LN | 41.3 | 33.5 | 32.3 | 32.2 | 38.6 | 34.3 | 35.4 | +10.5 | 0.22M |
| ● MLP | 37.2 | 39.3 | 40.3 | 34.4 | 46.2 | 37.8 | 39.2 | +14.3 | 108.62M |
| ● SAF | 38.8 | 31.7 | 31.9 | 30.5 | 38.0 | 32.2 | 33.9 | +9.0 | 1.77M |
| ● **Attn.** | **41.5** | **41.1** | **43.1** | **37.1** | **47.8** | **39.8** | **41.7** | **+16.8** | **40.75M** |

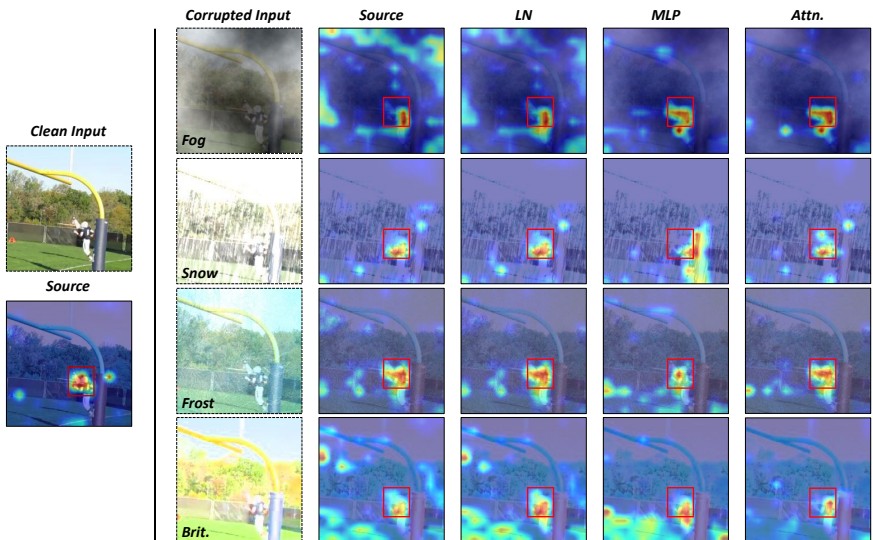

Figure 1: Grad-CAM visualization on ViT-based encoders with various modulation parameters. The comparisons are conducted with real-world weather corruptions, including fog, snow, frost, and brightness. For clear presentation, we introduce the **red box** to indicate the high activation region in the unbiased attention pattern, i.e., the discriminative semantic objective. Specifically, the label of the sample is *passing American football (in game)*, and the corresponding semantic objective is the running football player.

TTA methods in multi-modal scenarios. On the one hand, we discover that normalization statistics adaptation suffers from a challenge of *attention shifts* when dealing with biased modalities. As illustrated in Figure 1, the adapted model mistakenly focuses on biased and non-discriminative semantics (e.g., lawn, sky) when processing the sample with the label *passing American football (in game)*. In contrast, self-attention adaptation maintains a stable information flow on discriminative semantics (i.e., football player). In this paper, we argue that attention shifts are the primary factor preventing model generalization across domains. The results in Table 1 further corroborate the relationship between attention shifts and performance, where updating self-attention modules significantly outperforms other modulation parameters (e.g., LN, MLP). On the other hand, most methods focus on uni-modal adaptation and adapt modality-specific encoders independently, ignoring the synergy and complementarity in multi-modal adaptation. Although Shin et al. (2022) jointly consider multi-modal information to construct reliable pseudo-labels, this is established on the output space rather than the feature space, struggling to model complex modality associations. In conclusion, our work mainly focuses on two aspects: i) the attention shifts in distribution-shifted modalities and ii) the multi-modal synergy and complementarity.

From these observations, we develop a novel TTA approach oriented towards multi-modal models, called parameter-efficient attention transfer (PEAT). In the method, we counter attention shifts by performing adaptation on self-attention modules instead of layer normalizations. Motivated by Hu et al. (2022), the attention updates are learned in a smaller subspace via Low-Rank Adaptation (LoRA). Meanwhile, we propose a modality-aware LoRA method to introduce intra-modal and inter-modal interactions between cross-layer and cross-modality LoRAs. Specifically, our model use modality-specific parameters to compress the domain information of different modalities into a low-rank space, and then use modality-shared parameters to project them into a unified space, promoting feature alignment between modalities. In this way, we trade off parameter efficiency and performance, not only harnessing redundancy in LoRA parameters but also considering synergy and complementarity in multi-modal adaptation.

Our main contributions can be summarized as:

- We rethink the test-time adaptation (TTA) for multi-modal models, highlighting the *attention shifts* in distribution-shifted modalities, and the multi-modal synergy and complementarity.

- We propose a parameter-efficient attention transfer (PEAT) method for multi-modal TTA, obtaining a trade-off between performance and efficiency.

- We extend a vision-language benchmark dataset for multi-modal TTA and introduce eight text corruption types at character, word, and sentence levels. Extensive experiments with corruptions in various modalities, including video, image, audio, and text, demonstrate that PEAT improves the performance of entropy-based TTA method in multi-modal contexts.

## 2 RELATED WORK

**Test-Time Adaptation (TTA)** refers to domain adaptation in a source-free and online manner, which has been shown to boost model robustness against distribution shifts during testing. In the setting of fully TTA, the model must adapt given only the pre-trained parameters and the unlabeled test data. Tent (Wang et al., 2021), as the pioneer work, proposes to conduct adaptation on the affine parameters in normalization layers with entropy minimization. Since TTA proved effective as a general adaptation setting, subsequent works have expanded its practicality in a broader range of contexts, including, but not limited to, i) robust adaptation (Niu et al., 2022; 2023; Yuan et al., 2023; Lim et al., 2023) for practical scenarios involving mixed domain shifts, small batch sizes, label shifts, etc.; ii) continual adaptation (Wang et al., 2022; Lee et al., 2024a) against continually changing distribution shifts, which suffers from error accumulation and catastrophic forgetting; iii) efficient adaptation (Song et al., 2023; Niu et al., 2024) for resource-limited on-device learning.

However, these works are constrained to uni-modal adaptation, demonstrating suboptimal improvements when applied to multi-modal models. A recent work called READ (Yang et al., 2024) attributes it to multi-modal reliability bias and proposes self-adaptive attention-based fusion (SAF) to mitigate the dominance of biased modalities. Compared to READ, our work has the following key differences. i) Motivation differences. READ focuses on solving multi-modal reliability bias through self-adaptive fusion. However, this work is based on the observation that normalization adaptation methods (e.g., SAR) struggle with attention shifts; thus, we aim to find more effective adaptation parameters. ii) Modulation parameter differences. We propose the attention transfer method to dynamically learn cross-domain attention patterns in modality encoders rather than in the fusion module. iii) Updating method differences. READ directly updates the attention-based fusion module. Our work proposes a low-rank adaptation method with intra-modal and inter-modal interactions, which reduces the number of tunable parameters and considers the multi-modal association.

**Parameter-Efficient Fine-Tuning (PEFT)** is a technique designed to adapt large pre-trained models for specific downstream tasks without requiring full retraining. PEFT updates only a small subset of additional parameters while freezing the majority of the model's structure, offering significant advantages in terms of efficiency, accessibility, and adaptability. The promising advances in this sphere include adapter-based methods (Rebuffi et al., 2017; Houlsby et al., 2019), LoRA-based methods (Hu et al., 2022; Guo et al., 2024a), prompt-based methods (Lester et al., 2021), and many other variants (Li & Liang, 2021; Ben Zaken et al., 2022). Among these, LoRA-based methods have achieved a trade-off between parameter efficiency and performance, attracting widespread attention in practical applications. However, recent research observes the redundancy of LoRA parameters.

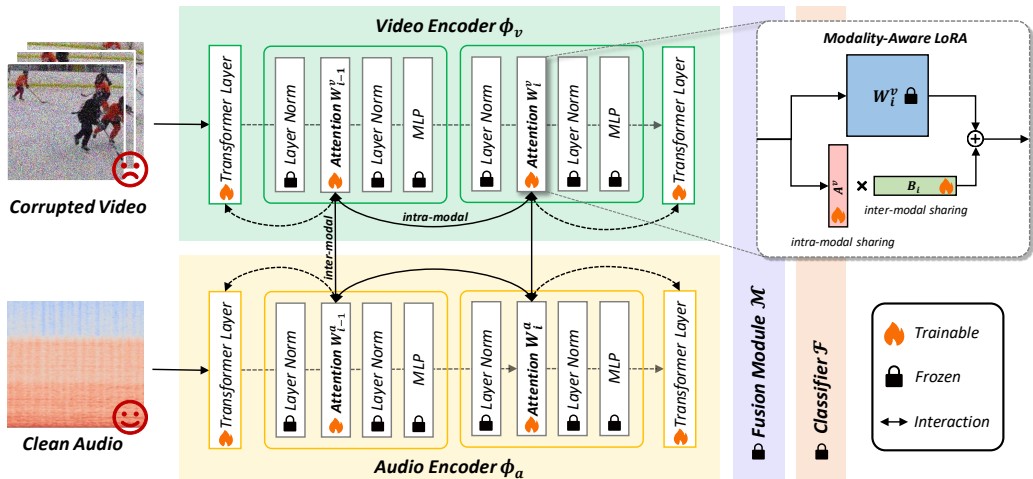

Figure 2: The illustration of Parameter-Efficient Attention Transfer (PEAT). During adaptation, the corrupted modalities are input into corresponding modality-specific encoders, and the output embeddings are concatenated at the token level for fusion. In the method, we update self-attention projections both in the encoders and the fusion module using Low-Rank Adaptation (LoRA). To further achieve multi-modal collaboration, we introduce the Modality-Aware LoRA method with intra-modal interactions (*i.e.*, $\mathbf{A}^a$ and $\mathbf{A}^v$) and inter-modal interactions (*i.e.*, $\mathbf{B}_i$).

VB-LoRA (Li et al., 2024) replaces low-rank matrices with a shared vector bank. RaSA (He et al., 2025) demonstrates that sharing ranks across layers leads to lower reconstruction error and thus better expressive capacity. These findings collectively suggest that LoRA parameters have not been fully utilized and that different LoRAs exhibit similarities across layers and modules. In this paper, we further explore the redundancy in LoRA parameters and introduce multi-modal interactions using shared low-rank matrices.

# 3 METHODOLOGY

## 3.1 PRELIMINARIES

**Problem Definition.** Without loss of generality, we consider multi-modal TTA within the context of an audio-video classification task. Specifically, we utilize the most popular architecture in multi-modal models, featured with modality-specific encoders and a modality-unified fusion module, which can be denoted as $f(\cdot) = \{\phi_a(\cdot), \phi_v(\cdot), \mathcal{M}(\cdot), \mathcal{F}(\cdot)\}$, where $\phi_a(\cdot)$ and $\phi_v(\cdot)$ refer to the transformer encoders for audio and video modality, $\mathcal{M}(\cdot)$ and $\mathcal{F}(\cdot)$ represent the fusion module and the following classification head.

Domain adaptation aims to transfer the model from the source domain $P(\mathbf{x})$ to the target domain $Q(\mathbf{x})$, where $P(\mathbf{x})$ and $Q(\mathbf{x})$ have a large distribution gap. Before adaptation, the base model $f_\theta(\cdot)$ parameterized with $\theta$ has been pre-trained on the labeled data $\mathcal{D}_{source} = \{(\mathbf{x}_i, y_i)\}_{i=1}^N$, where the multi-modal input $\mathbf{x}_i = \{x_i^a, x_i^v\} \sim P(\mathbf{x})$ consists of audio $x_i^a$ and video $x_i^v$ pairs. $f_\theta(\cdot)$ can exhibit excellent inference performance on the in-distribution (ID) test samples drawn from $P(\mathbf{x})$ but struggles to generalize to out-of-distribution (OOD) samples $\mathcal{D}_{target} = \{\mathbf{x}_i\}_{i=1}^M \sim Q(\mathbf{x})$. In this paper, we propose a parameter-efficient adaptation method using low-rank adaptation (LoRA).

**LoRA** models the incremental update of a pre-trained weight matrix $\mathbf{W}_0 \in \mathbb{R}^{d_1 \times d_2}$ by the product of two low-rank matrices $\mathbf{A} \in \mathbb{R}^{d_1 \times r}$ and $\mathbf{B} \in \mathbb{R}^{d_2 \times r}$, where $r \ll \{d_1, d_2\}$. For $\mathbf{h} = \mathbf{W}_0\mathbf{x}$, the modified forward pass is

$$\mathbf{h} = \mathbf{W}_0\mathbf{x} + \Delta\mathbf{W}\mathbf{x} = \mathbf{W}_0\mathbf{x} + \mathbf{A}\mathbf{B}^\top\mathbf{x}. \tag{1}$$

In practical applications, the matrix $\mathbf{A}$ is initialized with a random Gaussian distribution and $\mathbf{B}$ with zeros, setting the initial $\Delta\mathbf{W}$ to zero for training.

## 3.2 Parameter-Efficient Attention Transfer

**Motivating Observation.** Transformer has emerged as the most popular architecture for scaling in multi-modal learning. Therefore, we highlight the exploration of the adaptability of transformer-based models. The mainstream TTA methods (Niu et al., 2023) tend to mitigate covariance shift with layer normalization (LN) adaptation. Recent work (Yang et al., 2024) has introduced self-adaptive attention-based fusion (SAF) against multi-modal reliability bias. In order to determine the most effective modulation parameters, we conduct empirical studies on various transformer components, including LN, MLP, SAF, and self-attention. It is worth noting that, to ensure a fair comparison, all experiments were performed with the same unsupervised objectives, avoiding collapse during the adaptation process. As demonstrated in Table 1, self-attention adaptation exhibits significant superiority compared to other modulation parameters. In the transformer architecture, the attention map $\mathbf{softmax}(QK^\top)$ serves as the only way for information to flow between tokens. For further discussion, we explore the attention map in transformer layers. According to visualization results in Fig. 1, the source model mistakenly focuses on biased and non-discriminative semantics (e.g., lawn, sky) under distribution shifts. Consistent with the quantitative experiments, LN adaptation fails to improve the attention patterns of the source model, while self-attention adaptation captures the class-related objective (i.e., football player) from distribution-shifted contexts. In this paper, we refer to this phenomenon of misfocus or wrong activation in attention maps as *attention shifts*. Based on the above observations, we believe that attention shifts impair the expressiveness of biased modalities, leading to information discrepancies across modalities. To address this, we propose optimizing the biased attention pattern during testing by leveraging a signal derived from entropy minimization.

**Attention Transfer.** The *attention shifts* illustrate the limitation of LN adaptation (e.g., SAR (Niu et al., 2023)) for transformer-based models. Therefore, we propose a novel approach of *attention transfer*, which aims to dynamically learn attention patterns across domains. Specifically, we conduct self-attention adaptation in modality-specific encoders $\phi_a(\cdot)$ and $\phi_v(\cdot)$, while keeping the fusion module $\mathcal{M}(\cdot)$ and the classification head $\mathcal{F}(\cdot)$ frozen. In this way, the model can maintain a stable information flow on the most discriminative content, promoting the alignment of multi-modal representations from the target domains to the source domain.

Notably, self-attention adaptation is an effective but not efficient method. In transformer layers, the embedding dimension is donated as $d_{\mathrm{model}}$, then a layer normalization comprises $d_{\mathrm{model}} \times 2$ parameters, while an attention module encompasses $(d_{\mathrm{model}} + 1) \times d_{\mathrm{model}} \times 3$ parameters. The number of parameters in self-attention modules is significantly larger than that of layer normalizations. To address this, we develop a parameter-efficient attention transfer method via low-rank adaptation (LoRA). Specifically, following Eq. 1, the low-rank updates $\Delta\mathbf{W}$ are incorporated into the query $\mathbf{W}_Q$, key $\mathbf{W}_K$ and value $\mathbf{W}_V$ projection matrices in the self-attention module. In this way, the number of parameters required can be decreased to $d_{\mathrm{model}} \times r \times 4$. Since $r \ll d$, it has a comparable number of parameters to LN. The $\Delta\mathbf{W}$ in multi-modal models can be decomposed as:

$$\Delta\mathbf{W}_i^a = \mathbf{A}_i^a \mathbf{B}_i^{a\top}, \Delta\mathbf{W}_i^v = \mathbf{A}_i^v \mathbf{B}_i^{v\top}, i \in [0, D_s), \tag{2}$$

where $D_s$ denotes the number of modality-specific layers in encoders.

One intuitive observation is that, when a multi-modal model suffers from distribution shifts, its robustness usually depends on the reliable modality information. READ (Yang et al., 2024) is motivated by this insight and propose self-adaptive attention-based fusion (SAF) to make the model focus on unbiased modalities. However, this method cannot improve the degraded modality representations, leading to a low information gain. Thus, our method proposes to leverage cross-modal information to generate reliable modality representations. Toward this goal, we introduce intra-modal and inter-modal interactions between LoRAs, taking into account synergy and complementarity between multiple modalities. Specifically, we achieve the interactions by shared low-rank matrices while reducing the redundancy in LoRA parameters.

**Intra-Modal Interaction** aims to construct domain information that is independent between modalities but consistent within modalities. During inference, the distribution-shifted modality input is processed by a sequence of transformer layers. The shared domain knowledge ensures a consistent understanding of domain-specific features, maintaining a stable and effective information flow in the model. In the method, the matrices $\mathbf{A}_i^a$ and $\mathbf{A}_i^v$ in Eq. 2 are shared in each modality encoder, which can be uniformly expressed as $\mathbf{A}_a$ and $\mathbf{A}_v$, i.e., $\mathbf{A}_0^a = \mathbf{A}_1^a = \cdots = \mathbf{A}_D^a = \mathbf{A}^a$ and $\mathbf{A}_0^v = \mathbf{A}_1^v = \cdots = \mathbf{A}_D^v = \mathbf{A}^v$.

**Inter-Modal Interaction** aims to model semantic consistency across modalities. In real-world scenarios, different modalities are often closely related. Taking audio-visual learning as an example, both the barking sound of the dog and its appearance can be related to the concept of "dog", which is called *semantic consistency*. In multi-modal adaptation, this cross-modal consistency allows biased modalities to be enhanced by reliable information from other modalities. To establish a bridge between modalities, the low-rank matrices $\mathbf{B}_i^a$ and $\mathbf{B}_i^v$ in Eq. 2 share parameters layer by layer between modality-specific encoders, which can also be uniformly expressed as $\mathbf{B}_i^s$, *i.e.*, $\mathbf{B}_i^a = \mathbf{B}_i^v = \mathbf{B}_i$.

In this way, modality-specific parameters compress the information of different modalities into a low-rank space, and then modality-shared parameters project them into a unified space, promoting cross-modal feature alignment.

### 3.3 SOURCE-AWARE ENTROPY MINIMIZATION

According to the setting of multi-modal TTA, attention transfer is performed in an unsupervised manner. To mitigate the negative transfer, the source domain knowledge is leveraged to provide regularization for the entropy minimization.

During adaptation, different samples produce various effects, and samples with high confidence are more valuable for domain transfer. Existing works (Niu et al., 2022; 2023) have utilized entropy as the confidence metric. Another related research (Lee et al., 2024b) focuses on the sensitivity to structural information, which is designed for visual modalities. In this paper, we attempt to employ source domain knowledge to judge the reliability of samples. Specifically, the samples that deviate far from the prior in the source domain are considered to have high uncertainty and are more likely to produce wrong gradients. Based on the assumption above, we propose to attach higher weights to samples with smaller divergence and lower entropy in optimization.

In the implementation, the Jensen-Shannon divergence ($\mathbf{D_{JS}}$) (Menéndez et al., 1997) of predictions measures the distance of each sample that deviates from the source domain. Formally, the sample-adaptive weight is given by:

$$w(\mathbf{x}) = \frac{1}{\exp[\mathbf{D_{JS}}(f_{\theta_t}(y|\mathbf{x}) \| f_{\theta_0}(y|\mathbf{x})) \cdot \mathbf{H}(f_{\theta_t}(y|\mathbf{x}))]} \tag{3}$$

where $f_{\theta_t}(y|\mathbf{x})$ denotes the softmax output of the adapted model at epoch $t$, $\mathbf{H}(\cdot)$ represents the information entropy. Correspondingly, $\theta_0$ is the parameters of the source model, which can be easily obtained from $\theta_t$ by disabling LoRA.

With the source-aware weight, the sample-wise loss of improved entropy minimization can be expressed as:

$$\mathcal{L}_{ent} = -w(\mathbf{x}) \sum_{y \in \mathcal{C}} f_{\theta_t}(y|\mathbf{x}) \log f_{\theta_t}(y|\mathbf{x}) \tag{4}$$

Furthermore, adaptation using entropy minimization tends to cause collapse, i.e., predicting all samples to a single class (Niu et al., 2023), which attributes to the unbalanced label distribution. Thus, a diversity-promoting term $\mathcal{L}_{div} = \sum_{y \in \mathcal{C}} \hat{p}_c \log \hat{p}_c$ is introduced following preliminary works (Liang et al., 2020), where $\hat{p}_c = \frac{1}{B} \sum_{i=1}^{B} f_{\theta_t}(y|\mathbf{x}_i)$ is the average of softmax output of test samples in each mini-batch of size $B$, and $\mathcal{C}$ is the model output space.

## 4 EXPERIMENTS

**Datasets and Models.** Previous research (Yang et al., 2024) has constructed two audio-visual benchmark datasets, Kinetics (Kay et al., 2017) and VGGSound (Chen et al., 2020), and introduced 15 types of video corruptions and 6 types of audio corruptions. The ViT-based CAV-MAE (Gong et al., 2023) model serves as the source model, which is pre-trained on web-scale audio-visual data and fine-tuned on the training sets of Kinetics50 and VGGSound dataset.

To further verify the applicability of multi-modal TTA methods, we provide a vision-language dataset, UPMC-FOOD101 (Wang et al., 2015), as a novel benchmark and introduce 8 types of text corruptions at the character, word, and sentence levels. We use the pre-trained bert-base-uncased (Devlin et al., 2019) model to extract text features and use pre-trained ViT (Kolesnikov

Table 2: Comparison with state-of-the-art methods on Kinetics-C with corrupted video modality (severity level 5) regarding **Accuracy (%, ↑)**.

| Method | Noise | | | Blur | | | | Weather | | | | Digital | | | | Avg. |
|---|---|---|---|---|---|---|---|---|---|---|---|---|---|---|---|---|
| | Gauss. | Shot | Impul. | Defoc. | Glass | Mot. | Zoom | Snow | Frost | Fog | Brit. | Contr. | Elas. | Pix. | JPEG | |
| Source | 46.9 | 48.5 | 46.9 | 67.4 | 62.1 | 71.5 | 66.7 | 61.3 | 61.2 | 46.6 | 75.3 | 52.0 | 66.2 | 66.5 | 62.2 | 60.1 |
| • Tent (ICLR'21) | 46.3 | 47.4 | 46.4 | 67.5 | 62.6 | 71.4 | 67.8 | 61.7 | 61.8 | 37.6 | 75.4 | 51.2 | 67.3 | 67.6 | 62.9 | 59.7 |
| • EATA (ICML'22) | 46.9 | 48.2 | 47.0 | 67.6 | 63.4 | 71.4 | 67.8 | 62.2 | 62.3 | 47.2 | 75.3 | 52.1 | 66.9 | 67.4 | 63.2 | 60.6 |
| • SAR (ICLR'23) | 47.0 | 48.4 | 47.1 | 67.4 | 62.1 | 71.5 | 66.8 | 61.3 | 61.1 | 46.5 | 75.4 | 52.1 | 66.2 | 66.5 | 62.4 | 60.1 |
| • READ (ICLR'24) | 48.9 | 49.9 | 48.7 | 67.8 | 65.0 | 71.7 | 68.8 | 64.0 | 64.5 | 55.2 | 75.5 | 53.4 | 68.2 | 68.2 | 65.0 | 62.3 |
| • TSA (ICML'25) | 52.6 | 52.3 | 52.0 | 68.7 | 68.0 | 70.7 | 68.8 | 65.2 | 66.6 | 64.3 | 74.6 | 57.4 | 70.5 | 69.0 | 66.2 | 64.5 |
| • **Ours** | **51.4** | **51.8** | **50.6** | **70.5** | **70.8** | **73.8** | **72.4** | **67.6** | **68.2** | **66.7** | **75.9** | **58.9** | **73.5** | **72.7** | **70.4** | **66.4** |

Table 3: Comparison with state-of-the-art methods on Kinetics-C (left) and VGGSound-C (right) with corrupted audio modality (severity level 5) regarding **Accuracy (%, ↑)**.

| Method | Noise | | | Weather | | | Avg. | Noise | | | Weather | | | Avg. |
|---|---|---|---|---|---|---|---|---|---|---|---|---|---|---|
| | Gauss. | Traff. | Crowd. | Rain | Thund. | Wind | | Gauss. | Traff. | Crowd. | Rain | Thund. | Wind | |
| Source | 74.0 | 65.5 | 67.9 | 70.4 | 67.9 | 70.3 | 69.3 | 37.2 | 21.2 | 16.8 | 21.6 | 27.3 | 25.5 | 24.9 |
| • Tent | 73.9 | 67.2 | 69.2 | 70.4 | 66.5 | 70.6 | 69.6 | 11.6 | 3.0 | 1.9 | 3.1 | 5.9 | 4.2 | 4.9 |
| • EATA | 73.8 | 67.0 | 69.0 | 70.6 | 69.0 | 70.5 | 70.0 | 40.3 | 27.5 | 26.0 | 28.5 | 35.3 | 31.4 | 31.5 |
| • SAR | 73.7 | 65.8 | 68.3 | 70.5 | 68.1 | 70.2 | 69.4 | 38.0 | 8.9 | 8.6 | 14.5 | 28.3 | 18.1 | 19.4 |
| • READ | 74.4 | 68.8 | 69.8 | 71.2 | 71.6 | 70.6 | 71.0 | 40.3 | 29.1 | 26.9 | 30.8 | 36.5 | 30.7 | 32.4 |
| • TSA | 74.5 | 69.6 | 70.5 | 71.4 | 72.0 | 71.0 | 71.5 | 41.5 | 31.8 | 30.9 | 32.6 | 38.9 | 32.6 | 34.7 |
| • **Ours** | **75.5** | **72.5** | **73.7** | **72.6** | **75.6** | **72.9** | **73.8** | **42.7** | **40.7** | **42.2** | **37.9** | **47.7** | **39.9** | **41.9** |

et al., 2021) on ImageNet to extract image features. Meanwhile, we use transformer encoders as modality encoders and the fusion module. The model is trained on the training sets of UPMC-FOOD101, obtaining the corresponding source models.

In the experiments, the clean datasets are the source domain, and the corrupted datasets are the target domain. As a result, we obtain the Kinetics50-C and VGGSound-C benchmarks with either corrupted audio or corrupted video modalities and the UPMC-FOOD101-C benchmark with either corrupted text or corrupted image modalities. Each type of corruption has five levels of severity. In order to check the performance under the worst corruption case, we focus on testing with corrupted data of high severity level.

**Compared Methods.** To evaluate the proposed method, we conduct contrast experiments with the following state-of-the-art (SOTA) methods, which involve uni-modal TTA and multi-modal TTA. Tent (Wang et al., 2021), EATA (Niu et al., 2022), and SAR (Niu et al., 2023) are representative entropy-based methods for uni-modal TTA, while READ (Yang et al., 2024) and TSA (Chen et al., 2025) are designed for multi-modal TTA tasks.

**Implementation Details.** For test-time adaptation, we update parameters using the Adam optimizer, with a batch size of 64 for audio-visual benchmarks, 128 for vision-language benchmark, and a learning rate of 0.0001 for all benchmarks. Additionally, we initialize trainable LoRA matrices with Kaiming uniform initialization (He et al., 2015), with $r = 16$ and $\alpha = 16$ for audio-visual benchmarks and $r = 4$ and $\alpha = 4$ for the vision-language benchmark.

## 4.1 COMPARISON WITH PREVIOUS METHODS

**Results under Uni-Modal Distribution Shifts.** Uni-modal distribution shifts have been discussed in the recent multi-modal TTA works (Yang et al., 2024; Chen et al., 2025). For a comprehensive evaluation, we conduct experiments under two challenging settings: i) distribution shifts of high severity ii) mixed distribution shifts.

*i) Distribution Shifts of High Severity:* To highlight the effectiveness and robustness, the following comparison focuses on the challenging scenarios where the dominant modalities suffer from high-severity corruptions. Notably, the dominant modalities of the Kinetics-C benchmark, VGGSound-C benchmark, and UPMC-FOOD101-C benchmark are video, audio, and text modalities, respectively.

Table 4: Comparison with state-of-the-art methods on UPMC-FOOD101-C with corrupted text modality (severity level 5) regarding **Accuracy (%, ↑)**.

| Method | Character | | | Word | | | Sentence | | Avg. |
|---|---|---|---|---|---|---|---|---|---|
| | Insert | Replace | Delete | Synon. | Split | Delete | Exten. | Trans. | |
| Source | 54.6 | 53.6 | 54.9 | 87.4 | 77.7 | 75.0 | 74.5 | 69.2 | 68.3 |
| • Tent | 54.4 | 53.7 | 55.2 | 87.5 | 78.0 | 75.1 | 75.6 | 69.5 | 68.6 |
| • EATA | 55.8 | 54.7 | 55.4 | 87.4 | 77.9 | 75.1 | 75.4 | 69.5 | 68.9 |
| • SAR | 55.2 | 54.3 | 55.5 | 87.5 | 78.0 | 75.2 | 75.7 | 69.5 | 68.9 |
| • READ | 55.8 | 54.6 | 55.3 | 87.4 | 77.7 | 75.0 | 74.4 | 69.4 | 68.7 |
| • **Ours** | **59.6** | **58.1** | **57.7** | **87.5** | **78.5** | **75.2** | **80.1** | **70.0** | **70.8** |

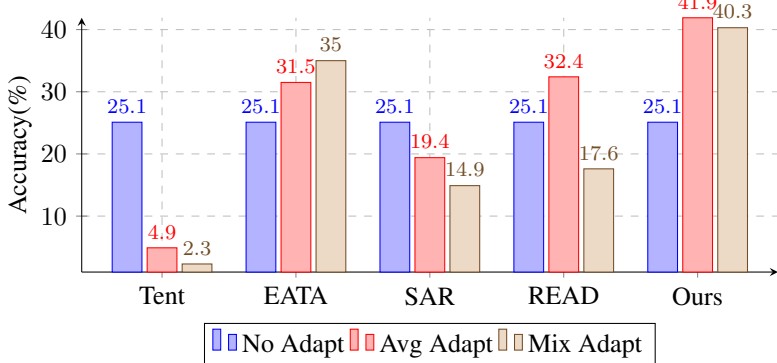

Figure 3: Performance of TTA methods under the mixture of 6 different audio corruption types (VGGSound-C).

The comparison results on the three benchmarks are reported in Tables 2, 3 and 4, where our PEAT method has shown significant superiority compared to existing methods when dealing with various datasets and various modality corruptions. Compared to the previous SOTA methods, we achieve an average performance improvement of 9.5% on the benchmark with audio corruptions, while 4.1% and 2.1% on the benchmarks of audio and text corruptions, respectively. Meanwhile, PEAT has been shown to achieve a balance between parameter efficiency and performance, which delivers a significant improvement over LN adaption while only requiring a comparable number of parameters.

*ii) Mixed Distribution Shifts:* We evaluated the performance on a mixture of 6 audio corruption types at severity levels 5 on the VGGSound-C benchmark. According to Figure 3, the mix adapt accuracy of other methods is significantly lower than the average adapt accuracy across different severity levels. In contrast, EATA and PEAT provide stronger robustness against various distribution shifts.

**Results under Multi-Modal Distribution Shifts.** Existing methods only consider the mild setting under unimodal corruptions. It will be more challenging to handle multi-modal TTA tasks suffering from multiple distribution-shifted modalities. Thus, we explore the multi-modal distribution shifts on the audio-visual benchmark. For instance, in a snowy environment, heavy snow and strong winds always occur together, which causes distribution shifts both in the

Table 5: Comparison with state-of-the-art methods on VGGSound-C with corrupted audio and video modality (severity level 5) regarding **Accuracy (%, ↑)**.

| Method | Snow&Wind | Fog&Rain | Frost&Traff. | Brit.&Crowd. | Avg. |
|---|---|---|---|---|---|
| Source | 8.8 | 5.0 | 8.9 | 9.1 | 7.9 |
| • Tent | 1.4 | 0.7 | 1.1 | 1.1 | 1.1 |
| • EATA | 10.4 | 4.9 | 9.3 | 11.4 | 9.0 |
| • SAR | 4.1 | 2.3 | 3.2 | 3.9 | 3.4 |
| • READ | 13.3 | 11.9 | 15.0 | 17.2 | 14.3 |
| • Ours w/o SAEM | 23.0 | 23.6 | 26.2 | 35.6 | 27.1 |
| • **Ours** | **24.9** | **24.9** | **27.5** | **36.3** | **28.4** |

sampling of video and audio modalities. According to the experimental results in Table 5, we observe that our method achieves an average performance improvement of 14.1% under multi-modal distribution shifts, which proves that our method leverages reliable modality information to improve the quality of modality representations.

Table 6: Ablation studies on VGGSound-C with corrupted audio modality (severity level 5) regarding **Accuracy (%)**. All methods are compared in the context of attention transfer.

| Method | Noise | | | Weather | | | | |
| | Gauss. | Traff. | Crowd. | Rain | Thund. | Wind | Avg. | Param. |
|---|---|---|---|---|---|---|---|---|
| Full Tuning | 41.7 | 41.3 | 43.2 | 37.7 | 48.1 | 40.4 | 42.1 | 38.98M |
| LoRA (baseline) | 41.5 | 41.0 | 42.8 | 37.3 | 47.6 | 39.7 | 41.7 | 1.08M |
| + intra-modal interaction | 41.2 | 40.9 | 42.6 | 37.2 | 47.6 | 39.9 | 41.6 | 0.84M (↓ 22%) |
| + inter-modal interaction | 41.5 | 41.0 | 42.8 | 37.0 | 47.9 | 39.6 | 41.6 | 0.43M (↓ 77%) |

## 4.2 Ablation Studies

**Effect of Parameter-Efficient Attention Transfer (PEAT).** The proposed method is a LoRA-based approach with inter-modal and intra-modal interactions. Therefore, we evaluate the contribution of each component in the architecture. From Table 6, we observe that the interactions based on parameter sharing maintain superior performance while reducing the number of parameters. Our ablation experiments demonstrate that multi-modal LoRA is cross-layer and cross-modal correlated, and we exploit this property to improve parameter efficiency. More ablation results are included in G.

**Effect of Source-Aware Entropy Minimization (SAEM).** SAEM attaches higher weights to high-quality samples that are closer to the source domain. This allows SAEM to prevent the model from overfitting to noisy samples and maintain the knowledge of source domain. Intuitively, the effect of SAEM is more obvious in stronger OOD scenarios. Therefore, we conduct ablation experiments under the setting of multi-modal corruption. The results in Table 5 exhibit that SAEM yields a significant accuracy improvement of 1.3%. This improvement proves that the source domain knowledge is a effective tool to measure the confidence of samples.

## 4.3 Efficiency Comparison

We conducted an efficiency analysis in terms of parameter count, memory usage, and inference speed. Since SAEM requires additional forward passes, it introduces an efficiency limitation and was therefore excluded from our evaluation. The results demonstrate that our method outperforms existing LN adaptation methods in accuracy while achieving higher efficiency. Compared to READ, which optimizes only the fusion

Table 7: Efficiency Comparison on VGGSound-C with corrupted audio modality (severity level 5).

| Method | Acc. (%) | Param. (M) | Mem. (G) | Throughput (Samples/s) |
|---|---|---|---|---|
| • Tent | 4.9 | 0.22 | 15.6 | 47.2 |
| • EATA | 31.5 | 0.22 | 27.5 | 45.9 |
| • SAR | 19.4 | 0.22 | 21.7 | 15.4 |
| • READ | 32.4 | 1.77 | **6.4** | 60.0 |
| • Ours ($N = 11$) | **41.6** | 0.43 | 14.0 | 49.0 |
| • Ours ($N = 5$) | 40.4 | 0.21 | 10.3 | 63.8 |
| • Ours ($N = 1$) | 36.6 | **0.06** | 6.8 | **70.4** |

layer, our method requires additional resources to update the modality encoders. However, as presented in Table 7, when the number of tunable layers $N$ is reduced, our method consistently delivers superior performance while achieving memory and computational efficiency comparable to READ.

## 5 Conclusion

In this paper, we explore test-time adaptation (TTA) methods oriented towards multi-modal models. Based on empirical studies, we reveal two key limitations of existing methods. Firstly, layer normalization (LN) adaptation struggles to handle the attention shifts when processing distribution-shifted modalities. Moreover, existing methods tend to adapt each modality independently, ignoring multi-modal synergy and complementarity. To address these issues, we propose a parameter-efficient attention transfer (PEAT) method, which adapts self-attention modules to dynamically learn cross-modal attention patterns. For efficiency, we learn attention updates in a smaller space via low-rank adaptation (LoRA). Furthermore, we decouple the low-rank matrices into intra-modal and inter-modal shared knowledge, associating reliable information across modalities. Extensive experimental results demonstrate the superiority of our PEAT method in terms of performance and robustness.

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

## A    REPRODUCIBILITY STATEMENT

We have already elaborated on all the models or algorithms proposed, experimental configurations, and benchmarks used in the experiments in the main body or appendix of this paper. Furthermore, we declare that the entire code used in this work will be released after acceptance.

## B    THE USE OF LARGE LANGUAGE MODELS

We use large language models solely for polishing our writing, and we have conducted a careful check, taking full responsibility for all content in this work.

## C    LIMITATIONS

Our work presents a parameter-efficient way to dynamically learn cross-domain attention patterns. We acknowledge that, as with any research endeavor, limitations exist in our work. Firstly, although the source-aware entropy minimization effectively mitigates overfitting on noisy samples, it requires an additional forward process, which limits the inference efficiency. Furthermore, while our method introduces inter-modal and intra-modal interactions by sharing the low-rank matrix between LoRAs, it does not fully exploit the potential of multi-modal association. Further utilizing multi-modal associations to enhance multi-modal test-time adaptation will be a promising approach.

## D    MORE DETAILS ABOUT THE BENCHMARKS

In this paper, we construct a new benchmark for multi-modal TTA upon the UPMC Food-101 Wang et al. (2015) dataset. UPMC Food-101 is a classification dataset that contains 90,704 image-text pairs and 101 classes, where the image and text pairs are noisy since all the images are obtained in an uncontrolled environment.

To explore adaptation to distribution shifts, we introduce different corruption types for image and text modalities. For images, we follow Hendrycks & Dietterich (2019) to apply 15 kinds of corruptions, and each corruption is with 5 kinds of severity levels for extensive validations. Specifically, the corruptions on image modality include "Gaussian Noise", "Shot Noise", "Impulse Noise", "Defocus Blur", "Glass Blur", "Motion Blur", "Zoom Blur", "Snow", "Frost", "Fog", "Brightness", "Elastic", "Pixelate", "Contrast", and "JPEG". Similar to the image modality, we design 8 kinds of text corruptions at the character, word and sentence levels. To be specific,

- *Character-level Corruptions* simulates typos in manual input or machine recognition, including insertion, replacement, and deletion. Given a character sequence $s = \{c_1, c_2, \cdots, c_n\}$, the result of insertion can be expressed as $s' = \{c_1, \cdots, c_k, \hat{c}, c_{k+1}, \cdots, c_n\}$, while replacement yields $s' = \{c_1, \cdots, c_{k-1}, \hat{c}, c_{k+1} \cdots, c_n\}$, and deletion gives $s' = \{c_1, \cdots, c_{k-1}, c_{k+1}, \cdots, c_n\}$, where $k \in [1, n]$ donates a random position number and $\hat{c}$ is a random letter.

- *Word-level Corruptions* evolve synonym replacement, splitting, and deletion. We assume that the word sequence can be expressed as $s = \{w_1, w_2, \cdots, w_n\}$. Synonym replacement preserves general meaning while replacing words at a random position $k$, which results in $s' = \{w_1, \cdots, w_{k-1}, \hat{w}, w_{k+1}, \cdots, w_n\}$, where $\hat{w}$ is one of the synonyms of $w_k$. Splitting means splitting words into smaller units, which can be expressed as $s' = \{w_1, \cdots, w_k[:u], w_k[u:], \cdots, w_n\}$, where $u$ is the split position in $w_k$. Deletion removes entire words, which yields $s' = \{w_1, \cdots, w_{k-1}, w_{k+1}, \cdots, w_n\}$.

- *Sentence-level Corruptions* explore distribution shifts dominated by semantic variation. Noise extension introduces irrelevant contexts $\tau$ into text $s$, which can be represented as $\{\tau_1, \cdots, s, \cdots, \tau_m\}$. Back-translation means translating the text into another language and then translating it back to the original language, where $s' = \mathrm{T}_{e2c}(\mathrm{T}_{c2e}(s))$, $\mathrm{T}_{e2c}$ and $\mathrm{T}_{c2e}$ refer to the translators from English to Chinese and from Chinese to English.

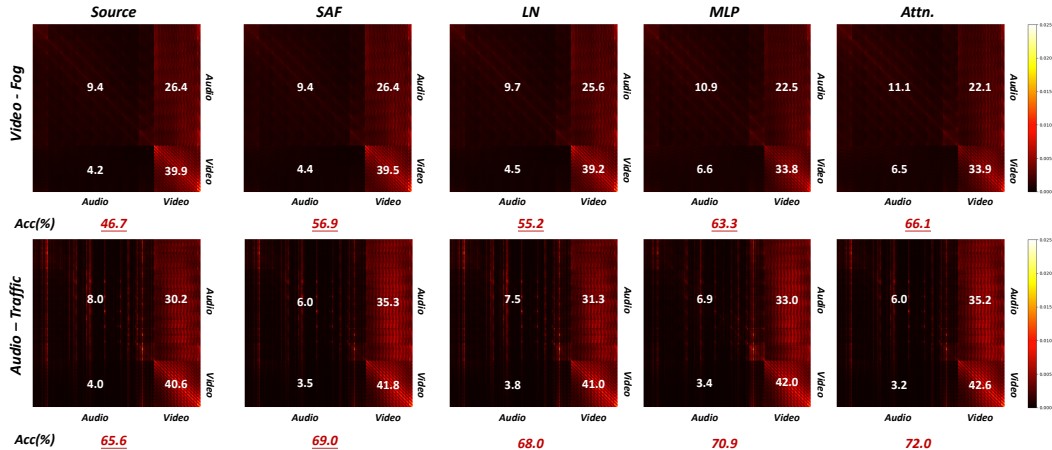

Figure 4: Visualization on the attention-based fusion module. The blocks of the top left and bottom right denote the self-attention between audios and videos, respectively. The blocks of the top right and bottom left denote the cross-attention from audio to video and video to audio, respectively. The number upon the blocks denotes the mean of attention values across the adaptation process, which is amplified by 10, 000 times for clarity.

## E    MORE VISUALIZATION RESULTS

In Figure 4, we exhibit the attention map in the fusion module and the number upon the blocks denotes the mean of attention values across the adaptation process. In the forward process, the concatenated audio and video token sequences $\mathbf{z} = \{z_1^a, z_2^a, \ldots, z_n^a, z_1^v, z_2^v, \ldots, z_m^v\}$ are fed into the fusion module. Thus, the corresponding attention map includes self-attention $M^{a2a}, M^{v2v}$ and cross-attention $M^{a2v}, M^{v2a}$, which can be expressed as:

$$M = \begin{bmatrix} M^{a2a} & M^{a2v} \\ M^{v2a} & M^{v2v} \end{bmatrix}$$

where the $x2y$ denotes the attention when modality $x$ is the query and modality $y$ is the key.

**How can we highlight the superiority of our method from the figure?** For example, the first row in the figure shows the attention maps when the video modality suffers from distribution shifts (i.e., fog). For robustness, the model should increase the importance of the unbiased modality, i.e., audio self-attention $M^{a2a}$ (top left) and video-to-audio cross-attention $M^{v2a}$ (bottom left). Correspondingly, $M^{v2v}$ and $M^{a2v}$ should be decreased. It is easy to observe that our method exhibits optimal attention scores compared to other methods.

## F    MORE EXPERIMENTAL RESULTS

Table 8: Comparison with state-of-the-art methods on UPMC Food-101-C with corrupted image modality (severity level 5) regarding **Accuracy (%, ↑)**.

| Method | Noise | | | Blur | | | | Weather | | | | Digital | | | | Avg. |
|---|---|---|---|---|---|---|---|---|---|---|---|---|---|---|---|---|
| | Gauss. | Shot | Impul. | Defoc. | Glass | Mot. | Zoom | Snow | Frost | Fog | Brit. | Contr. | Elas. | Pix. | JPEG | |
| Source | 82.8 | 82.2 | 82.6 | 84.6 | 84.3 | 85.4 | 83.0 | 82.2 | 84.3 | 81.5 | 89.2 | 81.2 | 88.1 | 88.1 | 88.0 | 84.5 |
| • Tent | 83.0 | 82.3 | 82.8 | 84.7 | 84.5 | 85.6 | 83.1 | 82.3 | 84.4 | 81.9 | 89.2 | 81.5 | 88.2 | 88.2 | 88.0 | 84.6 |
| • EATA | 83.1 | 82.5 | 82.9 | 84.7 | 84.5 | 85.6 | 83.2 | 82.4 | 84.5 | 82.0 | 89.2 | 81.6 | 88.2 | 88.2 | 88.0 | 84.7 |
| • SAR | 82.9 | 82.3 | 82.7 | 84.6 | 84.4 | 85.5 | 83.1 | 82.3 | 84.4 | 81.8 | 89.2 | 81.5 | 88.1 | 88.1 | 88.0 | 84.6 |
| • READ | 83.0 | 82.4 | 82.8 | 84.6 | 84.3 | 85.4 | 83.0 | 82.2 | 84.3 | 81.7 | 89.2 | 81.3 | 88.1 | 88.1 | 87.9 | 84.6 |
| • **Ours** | **84.0** | **83.6** | **83.9** | **85.3** | **85.2** | **86.0** | **83.9** | **83.4** | **84.8** | **83.2** | **89.3** | **82.8** | **88.4** | **88.5** | **88.2** | **85.4** |

Table 9: Continual TTA on VGGsound-C with corrupted audio modality (severity level 5). Each domain contains 1k samples.

| | | Noise | | | Weather | | | |
|---|---|---|---|---|---|---|---|---|
| Method | Setting | Gauss. | Traff. | Crowd | Rain | Thund. | Wind | Avg. |
| Source | - | 39.4 | 22.7 | 16.7 | 23.2 | 29.7 | 26.4 | 26.4 |
| Ours | reset each shift | 42.9 | 37.0 | 33.5 | 34.6 | 42.7 | 35.9 | 37.8 |
| Ours | continual | 42.9 | 35.0 | 37.9 | 34.7 | 45.6 | 37.3 | **38.9** |

## G    MORE ABLATION STUDY

**How does the LoRA rank influence the performance?**

Table 10: Performance comparison under different LoRA ranks on VGGSound-C with corrupted audio modality (severity level 5) regarding **Accuracy (%, ↑)**.

| | Noise | | | Weather | | | |
|---|---|---|---|---|---|---|---|
| $r$ | Gauss. | Traff. | Crowd. | Rain | Thund. | Wind | Avg. |
| 4 | 42.4 | 40.4 | 41.9 | 37.3 | 47.4 | 39.9 | 41.5 |
| 8 | 42.4 | 40.0 | 41.9 | 37.9 | 47.5 | 39.9 | 41.6 |
| 16 | 42.7 | 40.7 | 42.2 | 37.9 | **47.7** | 39.9 | 41.9 |
| 32 | 42.9 | 40.6 | **42.9** | **38.1** | **47.7** | **40.2** | **42.1** |
| 64 | **43.0** | **40.8** | 42.8 | 37.9 | **47.7** | 40.1 | 42.0 |

**How does the number of tunable layers influence the performance?**

We compared the performance and the number of parameters required when updating different numbers of layers. As the table illustrates, beyond updating the top three layers, additional layers provide only marginal improvements. Our interpretation is that lower layers typically focus on extracting generalizable low-level features, while higher layers are responsible for modeling more task-specific semantic features. Consequently, the self-attention mechanisms in these higher layers exhibit superior transferability.

Table 11: Ablation study for the number of tunable layers $D$ on VGGSound-C with corrupted audio modality (severity level 5) regarding **Accuracy (%, ↑)**.

| | | Noise | | | Weather | | | | |
|---|---|---|---|---|---|---|---|---|---|
| Method | $D$ | Gauss. | Traff. | Crowd | Rain | Thund. | Wind | Avg. | Param. (M) |
| Tent | - | 11.6 | 3.0 | 1.9 | 3.1 | 5.9 | 4.2 | 4.9 | 0.22 |
| READ | - | 40.3 | 29.1 | 26.9 | 30.8 | 36.5 | 30.7 | 32.4 | 1.77 |
| Ours | 1 | 41.3 | 35.5 | 36.1 | 34.0 | 41.2 | 35.6 | 37.3 | 0.06 |
| Ours | 2 | 42.2 | 36.8 | 38.0 | 35.7 | 43.7 | 37.3 | 38.9 | 0.10 |
| Ours | 3 | 42.6 | 38.1 | 39.7 | 36.6 | 44.9 | 38.5 | 40.1 | 0.14 |
| Ours | 4 | 42.6 | 38.9 | 40.5 | 37.4 | 46.2 | 39.1 | 40.8 | 0.17 |
| Ours | 5 | 42.8 | 39.0 | 40.7 | 37.3 | 47.0 | 39.4 | 41.0 | 0.21 |
| Ours | 11 | 42.7 | 40.7 | 42.2 | 37.9 | 47.7 | 39.9 | 41.9 | 0.43 |

