# OpenReview forum: "Parameter-Efficient Attention Transfer for Multi-Modal Test-Time Adaptation"
_ICLR.cc/2026/Conference — ICLR 2026 Conference Withdrawn Submission_

### Official Review · Reviewer_LWkx · 2025-10-23

**Soundness:** 2
**Presentation:** 2
**Contribution:** 2
**Rating:** 2
**Confidence:** 4

**Summary:**

This paper studies the problem of multimodal test-time distribution shift under the setting of multimodal test-time adaptation. The authors observe the attention shifts and insufficient exploitation of the synergy and complementarity among multiple modalities. Based on these, they propose Parameter-Efficient Attention Transfer (PEAT). Specifically, they adapt the self-attention modules and design a modality-aware low-rank adaptation method to dynamically learn cross-domain attention patterns. They conduct experiments Kinetics and VGGSound datasets.

**Strengths:**

1. The problem of multi-modal test-time adaptation is interesting and worth-exploring.
2. The authors have provided a detailed description of their proposed method.

**Weaknesses:**

1. The core idea of Parameter-Efficient Attention Transfer (PEAT) is not novel. The contribution is more incremental, combining existing techniques (e.g., LoRA, reweighting the loss function, cross-modal alignment) rather than introducing fundamentally new concepts.
2. For source-aware entropy minimization, the idea of using the source domain as regularization is not novel. Additionally, the use of Jensen-Shannon divergence is not sufficiently justified.
3. While there are empirical results, the paper lacks theoretical analysis (e.g., the theoretical advantages of source-aware entropy minimization, or theoretical bounds about attention transfer).
4. Multimodal test-time adaptation has been extensively studied recently, while the authors are mostly comparing with two of them (READ and TSA). There are many works that should be compared [1-5] but not sufficiently discussed or compared.
5. TSA is not compared in Figure 4&5.
6. The paper misses a thorough qualitative analysis, which is a must given the claims made by the authors. For instance, there is no examination of attention maps of the proposed method versus baselines.

[1] Analytic Continual Test-Time Adaptation for Multi-Modality Corruption

[2] Attention Bootstrapping for Multi-Modal Test-Time Adaptation

[3] Two-level test-time adaptation in multimodal learning

[4] Smoothing the shift: Towards stable test-time adaptation under complex multimodal noises

[5] Bridging the gap for test-time multimodal sentiment analysis

**Questions:**

Please refer to the weakness section.

---

### Official Review · Reviewer_cmq9 · 2025-10-29

**Soundness:** 3
**Presentation:** 2
**Contribution:** 3
**Rating:** 4
**Confidence:** 4

**Summary:**

This paper introduces Parameter-Efficient Attention Transfer (PEAT), a new framework for multi-modal test-time adaptation (TTA). Unlike traditional TTA methods mainly designed for uni-modal settings, PEAT explicitly addresses two challenges in multi-modal adaptation: (i) attention shifts under distribution shifts, and (ii) insufficient modeling of cross-modal synergy. The method performs adaptation on self-attention modules using Low-Rank Adaptation (LoRA) to balance performance and efficiency. It further proposes a Modality-Aware LoRA design that enables both intra-modal and inter-modal knowledge sharing, and a Source-Aware Entropy Minimization (SAEM) objective to prevent overfitting on noisy samples. Experiments across audio-visual (Kinetics-C, VGGSound-C) and vision-language (UPMC-FOOD101-C) benchmarks demonstrate consistent improvements over state-of-the-art methods in robustness, efficiency, and cross-modal generalization.

**Strengths:**

1. Comprehensive experimental evaluation across multiple modalities and corruption types, showing significant improvements over previous approaches.
2. The paper is well-structured, with clear problem motivation, algorithmic explanation, and empirical justification

**Weaknesses:**

1. The novelty of the paper is questionable. The idea of adapting attention modules shares a highly similar motivation with READ, making the contribution appear incremental rather than fundamentally new. Moreover, Table 6 does not clearly demonstrate how the proposed approach outperforms standard LoRA beyond efficiency considerations. It seems that the core novelty, namely the parameter-efficient attention transfer, does not provide substantial performance gains. The comparison in Table 6 should also include results from conventional LN adaptation and the attention-fusion method used in READ for a fair evaluation.
2. The proposed robust entropy minimization (Equation 4) closely resembles the loss function in SAR, which also performs weighted entropy minimization. The paper does not clearly justify the benefit of introducing the Jensen–Shannon divergence term, nor does it provide any experimental evidence supporting its effectiveness. Both the main text and supplementary material lack ablations or visualizations to clarify this component’s contribution, which weakens the paper’s claimed novelty.
3. The paper does not specify how the number of tunable layers is chosen. From Table 7, when N=11, the efficiency advantage becomes unclear. It is not evident how the authors determine the trade-off between accuracy and efficiency, or how the parameter count scales with the number of adapted layers.
4. While the experiments are rich and comprehensive overall, the rationale behind introducing the Mixed Distribution Shifts setting is not clearly explained. The connection between mixed-shift scenarios and the paper’s central motivation of addressing attention shifts is unclear. Furthermore, on the UPMC-FOOD101-C dataset, there are no experiments involving audio corruption, even though both image and text corruptions are covered. This inconsistency in experimental coverage raises questions about the completeness of the evaluation.

**Questions:**

Please see the weaknesses

---

### Official Review · Reviewer_ABpT · 2025-10-31

**Soundness:** 2
**Presentation:** 3
**Contribution:** 2
**Rating:** 4
**Confidence:** 5

**Summary:**

This paper proposes a parameter-efficient fine-tuning method to address the challenges of multi-modal test-time adaptation. By incorporating a lightweight LoRA framework to adapt modality-aware attention layers, the method enables the model to fully exploit both intra-modal and inter-modal information, thereby enhancing adaptation performance. Furthermore, the authors introduce a novel source-aware loss function. The proposed algorithm is validated through various experiments under both single-modality and simultaneous dual-modality corruption scenarios.

**Strengths:**

The experimental validation in this paper is a significant strength.

(1).The authors conduct extensive experiments that cover not only single-modality corruptions but also more challenging cases of simultaneous dual-modality corruptions. This demonstrates the method's robustness in complex, real-world settings.

(2).The introduction of a new vision-language benchmark (based on UPMC-FOOD101) is a valuable contribution. It effectively shows that the proposed method is not limited to audio-visual tasks and can be generalized to other multi-modal pairings.

(3).The paper includes a meticulous analysis of key factors such as the number of tunable parameters, providing clear insights into the trade-offs between performance and efficiency. This level of detail greatly strengthens the credibility of the findings.

**Weaknesses:**

(1). The core contribution of this paper lies in applying a parameter-efficient fine-tuning method to the task of multi-modal test-time adaptation. However, the general idea of introducing lightweight, tunable modules to enhance model adaptation is not a particularly novel concept in the TTA literature. For instance, prior works such as [5] and [6] have employed methods like prompt, which are conceptually similar to the LoRA-based approach in this paper; both essentially add a small set of learnable parameters to improve performance on the target domain. The authors have not sufficiently articulated the unique advantages or motivations for choosing LoRA, especially for adapting self-attention layers, within the multi-modal TTA context. This lack of justification diminishes the perceived novelty of the proposed method.

(2). The paper suffers from notable omissions in its survey and citation of critical related work, particularly within the multi-modal TTA domain. The authors frame "attention shifts" and "multi-modal synergy" as the core motivations for their work in the abstract and introduction. However, these critical issues have already been investigated in the context of multi-modal TTA in prior work [1]. This paper fails to cite this highly relevant publication or compare its method against it, which weakens the claim of originality regarding the problem formulation.

(3). The literature review on multi-modal TTA in the "Related Work" section is not comprehensive. The authors have overlooked several important papers that use the same datasets (VGGSound-C, Kinetics-C) and similar or identical experimental setups. Consequently, the baseline methods selected for comparison are predominantly general-purpose, uni-modal TTA approaches, lacking a direct comparison with the most relevant state-of-the-art multi-modal TTA methods. It is strongly recommended that the authors cite and compare their work with relevant literature, such as [1-4].

(4). A significant and critical experiment is missing from the paper's evaluation: VGGSound-C with corrupted video modality (severity level 5). When its dominant visual modality is severely corrupted, the model is forced to rely on the often weaker audio signals for classification, creating a highly challenging test scenario. This particular experiment is crucial for validating the method's robustness when modality reliability drastically drops. Despite its importance, the results for this key experiment are not reported anywhere in the main paper or the appendix.

[1] Y. Z., et al. Attention Bootstrapping for Multi-Modal Test-Time Adaptation. AAAI 2025.

[2] M. C. et al. Test-Time Selective Adaptation for Uni-Modal Distribution Shift in Multi-Modal Data. ICML 2025.

[3] Z. G., et al. Smoothing the Shift: Towards Stable Test-Time Adaptation under Complex Multimodal Noises, ICLR 2025.

[4] H. D., et al. TOWARDS ROBUST MULTIMODAL OPEN-SET TEST-TIME ADAPTATION VIA ADAPTIVE ENTROPY-AWARE OPTIMIZATION. ICLR 2025.

[5] Y. Y., et al. Test3R: Learning to Reconstruct 3D at Test Time. Arxiv.

[6] Y. Z., et al. DPCore: Dynamic Prompt Coreset for Continual Test-Time Adaptation. ICML 2025.

**Questions:**

1. Several major concerns have already been detailed in the "Weaknesses" section above.

2. When motivating the paper's approach, you compare fine-tuning Layer Normalization (LN) layers with fine-tuning attention layers. However, the effectiveness of fine-tuning can vary significantly across different layers of a Transformer. As shown in prior work [1], in certain scenarios, adapting the mixed LN layers and attention layers can yield better results than adapting attention layers. Did you observe any such phenomena in your experiments? Could you clarify why attention adaptation was consistently superior in your specific multi-modal setting?

3. Could you elaborate on the motivation for introducing the Jensen-Shannon (JS) divergence into your source-aware loss function? What are the specific and significant advantages of using JS divergence compared to the original entropy minimization loss function used in READ [2], especially in the context of multi-modal TTA?


[1] J. L., et al. Two-Level Test-Time Adaptation in Multimodal Learning, ICLR Workshop 2024.

[2] M. Y., et al. Test-time adaption against multi-modal reliability bias. ICLR 2024.

---

### Note · Authors · 2025-11-14

I have read and agree with the venue's withdrawal policy on behalf of myself and my co-authors.